# Anderson–Fabry Disease: A New Piece of the Lysosomal Puzzle in Parkinson Disease?

**DOI:** 10.3390/biomedicines10123132

**Published:** 2022-12-05

**Authors:** Marialuisa Zedde, Rosario Pascarella, Francesco Cavallieri, Francesca Romana Pezzella, Sara Grisanti, Alessio Di Fonzo, Franco Valzania

**Affiliations:** 1Neurology Unit, Neuromotor and Rehabilitation Department, Azienda Unità Sanitaria Locale-IRCCS di Reggio Emilia, 42123 Reggio Emilia, Italy; 2Neuroradiology Unit, Radiology Department, Azienda Unità Sanitaria Locale-IRCCS di Reggio Emilia, 42123 Reggio Emilia, Italy; 3Neurology Unit, Stroke Unit, Dipartimento di Neuroscienze, AO San Camillo Forlanini, 00152 Rome, Italy; 4Clinical and Experimental Medicine PhD Program, University of Modena and Reggio Emilia, 41121 Modena, Italy; 5Neurology Unit, Foundation IRCCS Ca’ Granda Ospedale Maggiore Policlinico, 20122 Milan, Italy

**Keywords:** Anderson–Fabry disease, Parkinson disease, α-galactosidase, lysosomal enzymes, neurodegenerative, neuroimaging

## Abstract

Anderson–Fabry disease (AFD) is an inherited lysosomal storage disorder characterized by a composite and multisystemic clinical phenotype and frequent involvement of the central nervous system (CNS). Research in this area has largely focused on the cerebrovascular manifestations of the disease, and very little has been described about further neurological manifestations, which are known in other lysosomal diseases, such as Gaucher disease. In particular, a clinical and neuroimaging phenotype suggesting neurodegeneration as a putative mechanism has never been fully described for AFD, but the increased survival of affected patients with early diagnosis and the possibility of treatment have given rise to some isolated reports in the literature on the association of AFD with a clinical phenotype of Parkinson disease (PD). The data are currently scarce, but it is possible to hypothesize the molecular mechanisms of cell damage that support this association; this topic is worthy of further study in particular in relation to the therapeutic possibilities, which have significantly modified the natural history of the disease but which are not specifically dedicated to the CNS. In this review, the molecular mechanisms underlying this association will be proposed, and the available data with implications for future research and treatment will be rewritten.

## 1. Introduction

Anderson–Fabry Disease (AFD) is an inherited sphingolipidosis due to a deficit of the lysosomal enzyme alpha-galactosidase (a-GAL A), which is responsible of the hydrolysis of terminal, non-reducing α-D-galactose residues in α-D-galactosides, catalyzing many catabolic processes, including cleavage of glycoproteins, glycolipids, and polysaccharides. The a-GAL A is encoded by the *GLA* gene [1], located in the X chromosome at the position Xq22.1 [2]. AFD is an X-linked disease, affecting 1 in 40,000 males, but, unlike other X-linked diseases, the women carrying pathogenic *GLA* variants may have the same systemic involvement as men and with the same severity. Another issue which makes it hard to precisely estimate the prevalence of AFD is that the degree of loss of a-GAL A function due to a heterogeneous effect of *GLA* pathogenic variants may be responsible for two different clinical phenotypes: “classic” AFD and “late onset” AFD [3]. The loss of a-GAL A function is responsible for the accumulation of the main substrate, globotriaosylceramide (Gb3), in the lysosomes. The main sites of Gb3 accumulation are the endothelial cells and the smooth muscle cells of the vessel’s walls, the heart muscle and endocardial cells, the glomerular nephrons and CNS cellular populations (neurons and glial cells), as well as the skin and the eye. The sites of accumulation predict systemic involvement in AFD, and several organs are directly assessable by biopsy for confirming and grading Gb3 accumulation (i.e., the heart and kidney). By contrast, CNS tissues do not undergo biopsy for such reasons, and the information about CNS involvement in AFD is mostly indirect and relies on clinical events (transient ischemic attacks, cerebral hemorrhages, thrombosis, and lacunar infarcts) and on neuroimaging findings. The CNS pathology in AFD has mostly been considered a secondary manifestation of endothelial dysfunction, as suggested by evidence of accumulation of the glycosphingolipid globotriaosylceramide (Gb3/GL-3) in endothelial cells as a consequence of alpha-GAL A enzyme deficiency. Smooth cells also have been demonstrated to contribute to the pathogenesis by releasing GL-3 and sphingosyne1 phosphate. However, neuropathological studies are still insufficient, limiting the accuracy in assessing the causal relationship between the neurological phenotype and pathogenic mechanisms linked to AFD. CNS involvement in AFD is not uncommon, being reported in 34% of patients within a multispecialty screening study [3]. The prevalent involvement of the CNS refers to vascular damages of the parenchyma, especially the white matter, and the great vessels, e.g., vertebro-basilar dolichoectasia. Very little is known about pure neuronal and glial involvement in AFD. The therapies that have been available for a longer period of time, in particular enzyme replacement therapy (ERT), do not cross the blood–brain barrier, with an inevitable implication for the management of AFD-treated patients whose neurological involvement was not addressed. The more recently approved migalastat therapy has the potential to cross the blood–brain barrier, but its direct effects on the CNS have yet to be explored. Conversely, Parkinson disease (PD) is the most frequent neurodegenerative disease in the all-ages population [4,5], and its genetic determinants [6,7,8,9,10] are currently increasingly expanding the definition of monogenic causes and the role of specific genetic variants as predisposing factors on which a “second hit”, either genetic or non-genetic, could act as a precipitating event for the development of the disease [11,12,13,14]. Pathogenic variants of the GBA gene, which encodes for the lysosomal hydrolase glucocerebrosidase (GCase) and represent the most common genetic risk factor for PD in the population, fall into the latter group. The role of lysosomal dysfunction, including specific enzymatic alterations such as that of GCase, has been implicated in PD pathogenesis even in the absence of the corresponding gene mutations [15,16,17]. Recent reports suggested a possible association between AFD and PD. In particular, a higher prevalence of extrapyramidal signs and symptoms was observed in a subgroup of patients with AFD [18]. This association deserves further study and has a biological and pathophysiological rationale that has not yet been fully explored. The main purpose of this review is to describe the putative mechanisms of this association, the available data, and the possibilities for future development.

## 2. Anderson–Fabry Disease, Brain and Neurodegeneration

As with many diseases affecting the CNS, the diagnostic gold standard for neurological involvement in AFD is neuropathology assessment. However, the availability of brain tissue for diagnostic purposes is critical, especially in diseases such as the AFD that do not require neuropathological analysis for diagnosis. There are isolated reports of non-autoptic brain tissue investigations in ADF patients, e.g., in patients who underwent brain biopsy or neurosurgical interventions for tumors (e.g., meningiomas). There are also very few published data on autopsy series [19,20,21] of patients with AFD; such data were obtained mainly in a historical period prior to the introduction of ERT and without the aid of electron microscopy, which is usually used on biotic samples of other tissues (e.g., kidney, heart, and skin) for diagnostic purposes for AFD. Despite these limitations, the autopsy data provide us with some useful information to establish a solid basis for the pathophysiological hypotheses on the involvement of the CNS and its structures, from which the putative biological mechanisms of damage are better structured. One of the most interesting pieces of information coming from autoptic series [19] is that the storage of glycosphingolipids is documented in muscle cells of large, intermediate, and small cerebral vessels and in neuronal tissue, with a distribution involving mainly components of the limbic system (basolateral nuclei of the amygdala and supraoptic and paraventricular nuclei of the hypothalamus) and the brainstem structures (substantia nigra, pontine reticular formation, dorsal efferent nucleus of vagus, salivary nuclei, nucleus ambiguous, and mesencephalic nucleus of the fifth cranial nerve) but also the spinal cord and peripheral nervous system.

Despite this documented involvement of the midbrain structures and in particular of the substantia nigra, an extrapyramidal clinical expression had never been described in patients with AFD until recently. The reasons for this are probably to be found not only in the multifaceted phenotype of AFD, but also in the substantial modification of natural history after the introduction of ERT. Before the introduction of ERT, the involvement of heart and kidney was the main factor influencing the reduced life expectancy of AFD patients with the classic form, and CNS involvement was partially due to systemic damage. Conversely, after the introduction of ERT, the improved control of heart and kidney involvement and the increased life expectancy have reduced the burden of CNS damage secondary to systemic disease. On the other hand, the increasingly frequent identification of late onset variants made it easier to identify milder phenotypes, even on the neurological side.

In favor of the hypothesis of a primary neurological involvement in AFD, several studies based on longitudinal neuroimaging and neuropsychological assessments suggested a possible neurodegenerative pathogenesis not secondary to cerebrovascular disease. The 8-year follow-up study published by Lelieveld et al. [22] and focusing on neuropsychiatric symptoms and brain structural alterations in 14 AFD patients showed that, even without cognitive changes during the follow-up, a progressive decrease of hippocampal volume was recorded. Considering the not significant change of the white matter hyperintensities (WMHs) burden in the same interval, the loss of hippocampal tissue might be interpreted as pure neuronal involvement and therefore as a neurodegenerative phenotype of AFD patients, as suggested by the same research group at the baseline [23]. The lack of correlation between neuroimaging parameters and neuropsychiatric parameters is interpreted by the authors as successful compensation of the progressive hippocampal volume loss. The MRI finding of hippocampal atrophy is concordant with the already reported autoptic studies [24,25,26] demonstrating Gb3 storage in selective cortical and brainstem areas, including the neurons and ganglion cells of the hippocampus, in particular the presubiculum and the parahippocampal gyrus. It is interesting that in the case described by Okeda et al. [26] the dementia syndrome of the patients was probably due to extensive cerebrovascular involvement and not to the mild Gb3 storage in the presubiculum and parahippocampal gyrus. The Gb3 storage might cause functional damage to the cells through oxidative stress and energy metabolism compromise up to cell death. Moreover, the lack of correlation between WMH volume and hippocampal atrophy in AFD is another issue in favor of the hypothesis of pure neuronal involvement in AFD [23]. It seems that the hippocampal involvement could be considered a hallmark of neuronal involvement in AFD but without overt clinical manifestations outside of depression.

Neuroimaging studies [27] suggested that advanced techniques can help to define the pathophysiology of CNS involvement in FD outside of WMHs. In particular, the loss of brain tissue volume (brain atrophy, differentiated into white matter—WM—and grey matter—GM—atrophy) is visually assessed in conventional morphologic MRI sequences, but this qualitative assessment has obvious limitations in reproducibility and accuracy both in general and in AFD [28]. Advanced MRI techniques allow a quantitative and accurate assessment of brain volumes, and thus of atrophy. In AFD, as in other diseases, this evaluation is particularly relevant in absence of significant WMHs or cerebrovascular disease. Voxel-based morphometry (VBM) analysis has been applied to assess regional differences in GM volume in AFD patients, mostly failing to show a significant difference in comparison with healthy controls [29,30]. Conversely, atrophy in specific brain regions has been demonstrated in AFD, in particular in the thalami bilaterally and in the hippocampus [23,31], both with manual segmentation [23] and with VBM [31], suggesting direct neuronal involvement. An interesting finding is the significant reduction of the whole intracranial volume in AFD vs. healthy controls [31] with preserved fractional brain tissue volumes (GM, WM, and cerebrospinal fluid). This finding too was interpreted as the consequence of abnormal neural development.

Another advanced MRI technique has been applied to assess the microstructural WM changes, i.e., diffusion tensor imaging (DTI) [32]. The first studies focusing on cerebral diffusivity showed an elevated total brain parenchymal average diffusion constant in AFD vs. controls, and also in normal-appearing white matter [33], because of increased brain interstitial water content due to microvascular changes. A similar finding was found in frontal, parietal, and temporal normal-appearing WM by using ROI-based DTI analysis [34] and in the periventricular regions and the posterior portion of the thalamus by using voxel-based DTI study [35]. In the same line are the findings of tract-based spatial statistics (TBSS) analyses [36] and a combined TBSS–functional MRI (fMRI) study [37], in which in AFD patients without extensive WMH loads, extensive areas of reduced fractional anisotropy were identified in different supratentorial and infratentorial WM regions. All these data supported the presence of microvascular injury in the territory supplied by the long perforating arteries in an early stage, i.e. before WM lesions are detectable on conventional images.

A different approach is the use of fMRI to study the presence of possible functional changes in AFD patients. Gavazzi et al. [38] performed a motor task fMRI experiment (repetitive flexion–extension of the last four fingers of the right hand), finding an increased activation of additional cortical regions (cingulated motor area, secondary motor area, and primary sensorimotor cortex). Moreover, a resting-state fMRI (RS-fMRI) study [39] demonstrated an alteration of the corticostriatal pathway in AFD, with reduced functional connectivity (FC) between motor cortices and the caudate and lenticular nuclei, bilaterally, and between the left motor cortex and cerebellar areas. In the same direction, another study [37] demonstrated increased FC between the main hubs of the default mode network (DMN) (which is involved in the integration and coordination of sensorimotor and cognitive goal-directed activities) and different brain areas.

Finally, a promising technique is the qMRI, allowing a quantitative assessment of brain tissue relaxometry parameters and magnetic susceptibility; only isolated studies have applied it in AFD, but in one [39] of those papers, the focus was possible iron accumulation in the striatonigral pathway. Quantitative susceptibility mapping (QSM) may help to evaluate the pathologic tissue changes and, in particular, iron accumulation [40] in the striatonigral system due to different neurodegenerative disorders, including PD [41] and atypical Parkinsonism [42].

The application of advanced imaging techniques to study brain involvement in AFD suggested the presence of prodromal signs of neurodegeneration in AFD, with peculiar involvement of the motor system [18] in the form of a subclinical extrapyramidal phenotype [30] with motor slowing and postural instability more prevalent than rigidity and tremor. The above-reported neuroimaging findings supported the hypothesis of a neurodegenerative phenotype of AFD different from and not related to cerebrovascular involvement, which is better known and studied in the literature. This neurodegenerative phenotype was hypothesized also in the form of extrapyramidal dysfunction in general and mainly PD, as well as for other lysosomal storage disorders and in particular heterozygous GBA mutations.

## 3. Anderson–Fabry Disease and Extrapyramidal Phenotype

### 3.1. Literature Data

The link between PD and GBA mutations is well-documented, and it is one of the most relevant examples of the role of lysosomal dysfunction in the pathogenesis of PD [43]. Indeed, heterozygous GBA mutations are the most common genetic risk factors for PD without being able to cause Gaucher disease. The deficiency of other lysosomal enzymes is also associated with extrapyramidal disorders in general and PD in detail. Some examples of this association are sphingomyelin phosphodiesterase 1 (SMPD1) deficit causing Niemann–Pick disease A [44,45,46] and deficit of lysosomal integral membrane protein type 2 (LIMP-2), which is the glucocerebrosidase chaperone [47]. Interestingly both GCAse and LIMP-2 activity is impaired in the liver of patients with Niemann–Pick disease [48]. In this context, it is not surprising that there is a hypothesis of a link between PD and AFD, starting from the first findings of reduced α-Gal A activity in white blood cells of PD patients vs. controls [49]. In an experimental study [50] of a mouse model of AFD, the brain contained aggregates of alpha-synuclein, as in the human brains of PD patients. The association between AFD and PD was signaled more than 20 years ago through the publication of two cases [51,52] (a 68-year-old man and a 57-year-old woman), where the PD diagnosis was based upon the presence of a clinical extrapyramidal phenotype that was DOPA-responsive. The man received an autoptic confirmation of the diagnosis of AFD, and the woman received a molecular and genetic diagnosis (the R301P GLA mutation was identified). The first one had an atypical Parkinsonism, beginning at the age of 63 years, with axial signs (gait and postural instability), mild symmetrical rigidity, and pyramidal signs (mild right hemiparesis, generalized hyperreflexia, and positive Babinski sign on both sides). The authors reported the findings of brain neuroimaging (MRI) with multiple T2 hyperintensities in the basal ganglia and deep white matter regions. The woman described by Buechner et al. [52] had a mild Parkinsonism beginning at the age of 46 years, and the course was characterized by motor complications complicated by axial signs (gait impairment with freezing), although there was a good response to levodopa. This patient too had an extensive leukoencephalopathy with multifocal ischemic lesions, including in the head of the right caudate nucleus, on the brain MRI.

Lohle et al. [18] found that subjects with causal GLA mutations have a clinical bradykinetic motor phenotype, including slower gait and lower hand speed, but without the classic prodromal features of PD (hyposmia/anosmia, autonomic dysfunction, and REM sleep behavior disorder). This point was one of the main differences in comparison with individuals with GBA mutations [53].

Further attention on this issue was raised by a survey [54] proposed to a cohort of AFD patients with known GLA mutations, self-reporting the personal and familiar prevalence of PD. The survey had a wide diffusion because it was distributed within the AFD community through the National Fabry Disease Foundation and the Fabry Support and Information Group in the United States. The family history of PD was ascertained using a validated questionnaire [55], and the personal history was focused in already diagnosed PD and its treatment. The participating patients were also requested to report if any of their first-degree relatives showed common symptoms of PD (resting tremor, shuffling gait, stooped posture decreased arm swing, and rigidity). This was the same approach previously used to determine the penetrance of the most common forms of monogenic PD due to LRRK2 and GBA mutations [56,57]. Among 90 genetically confirmed AFD patients, 2 reported a previous diagnosis of PD (2/90, 2.2%), both >60 years old. The prevalence of PD diagnosis among participants >60 years old was 8.3% (2/24). The application of Kaplan–Meier survival analysis found an 11.1% age-specific risk of PD by age 70. The two AFD/PD patients harbored, respectively, the p.Y134X mutation and the p.E59V variant. The Y134X mutation is reported in association to a classical AFD phenotype [54,58,59], and the p.E59V variant was not previously reported, although a missense mutation in the same position of the GLA gene (E59K) [58] was described as causative of the classical AFD phenotype, and in silico prediction models [60] were in favor of the damaging potential of the variant. Regarding the family history, among 81 families, 4 (4.9%) had one first-degree family member with a previous PD diagnosis. This approach has several limitations, first of all the self-reporting itself and the lack of direct evaluation of the patient and the diagnostic pathway and follow-up, but it showed an association which needs further analysis, particularly in late onset AFD phenotypes due to specific GLA mutations.

This issue was addressed by Gago et al. [61], adding to the determination of the PD prevalence in AFD the description of clinical, biochemical, and vascular neuroimaging in AFD pedigrees with concomitant PD. The authors screened for PD on a clinical basis in 229 AFD patients with the same GLA mutation (p.F113L) belonging to 31 families. In this highly selected cohort, the prevalence of PD in AFD was 1.3% (3/229) (3% in patients aged ≥50 years). The adjusted prevalence of PD in the community-dwelling Portuguese population aged ≥50 years is 0.24% [62]. The three AFD patients with PD are a 73-year-old female and two 60- and 65-year-old males; they belong to three different pedigrees and presented akinetic–rigid PD, with a weak response to levodopa and dopaminergic deficiency on 18F-DOPA PET. They had no concomitant PD-related gene mutations in a dedicated panel. All patients have a severe AFD disease status in Mainz scores and a mild SVD pattern in brain MRI. One of the three patients had a worsening course with a moderate–severe Parkinsonism with very low response to levodopa and concomitant cerebrovascular lesions directly affecting nigrostriatal neural structures. The remaining two patients had a more prominent subcortical involvement but not basal ganglia lesions. This is not surprising because the p.F113L GLA mutation is associated with a high burden of cardiac and CNS manifestations [63,64]. It is possible that cerebrovascular burden may be associated to a different natural history of PD, as reported in sporadic PD with a more severe, axial phenotype of PD [65].

It is not possible to ascertain if this PD phenotype in late onset AFD is related to Gb3 deposition versus cerebrovascular lesions in the nigrostriatal network, but it is an intriguing hypothesis, and the simultaneous presence of the two mechanisms may be considered. More than one mechanism is involved in determining extrapyramidal phenotype in AFD patients, and it is possible that the concomitant involvement of the motor system, as suggested by advanced neuroimaging studies, may play an adjunctive role. A study [66] on 47 AFD patients vs. 49 healthy controls applied DTI to perform a diffusion MRI tractometry analysis focusing on the main afferent and efferent pathways of the cortico-striatal-thalamo-cortical loop. The authors found a microstructural involvement of the cortico-striatal-thalamo-cortical loop in FD patients, predominantly on the left side. The application of advanced neuroimaging techniques to the assessment of the morphometric and susceptibility changes of striatonigral pathway through the measurement of the volumes of different extrapyramidal relays and quantitative susceptibility mapping (QSM) was proposed by Russo C et al. [57] in 30 AFD patients (M/F = 11/19, mean age 42.6 ± 12.2) vs. 37 healthy controls (M/F = 16/21, mean age 43.2 ± 14.6). The main finding of this study was the increased susceptibility values of the SN and striatum in AFD vs. controls (respectively, *p* < 0.001 and *p* = 0.001) with reduced volume in the SN and not in the striatum; no difference was found in the other extrapyramidal structures. The documentation of the involvement of extrapyramidal networks and in particular SN in AFD supported the role of neurodegeneration as a mechanism for the subtle extrapyramidal phenotype emerging in the literature. Moreover, the regional pattern of susceptibility increase on QSM in the SN and striatum might support the prevalent akinetic/rigidity PD phenotype in AFD patients, as is already known in sporadic PD when the dentate nucleus is spared [67]. Another issue is the relative sparing of the striatum in volume in comparison to the SN, maybe because of a non-homogeneous susceptibility to neurodegeneration of the extrapyramidal structures. Indeed, in PD the pars compacta of the SN and nigrosome1 is involved earlier and more severely than the striatum [68,69,70]. The different patterns of involvement of nigral and striatal sub-components could explain the heterogeneous extrapyramidal clinical phenotypes also in AFD patients.

Recently, a single-patient study [71] with autoptic data was published on the association between AFD and Lewy pathology (LP) with extrapyramidal symptoms during life. The patient was a 58-year-old, cognitively preserved male AFD patient with predominant hypokinesia. The brain sections were evaluated by immunohistochemistry (CD77, a-synuclein, and collagen IV) and neuropathological staging. Severe neuronal loss in the substantia nigra pars compacta and LP corresponding to neuropathological stage 4 of Parkinson disease was seen without significant cerebrovascular damage or other diseases. Moreover, the different mechanisms underlying extrapyramidal phenotype in AFD and PD are supported by a small sample [72] of four AFD patients with extrapyramidal signs and without significant cerebrovascular burden on MRIs studied by using DaTSCAN. In all patients the functional study excluded the loss of pre-synaptic dopamine that is typical of PD. Finally, a recent single-case report [73] of a woman with early onset, dopamine-responsive PD and a bilateral nigrostriatal dysfunction on DaTSCAN do not support this hypothesis, but also the co-occurrence of AFD and classical PD by chance should not be neglected as a hypothesis in individual cases.

### 3.2. Biological Mechanisms

The reference model which has been used to interpret the biological mechanisms underlying the link between AFD and PD is the one provided by GBA and Gaucher disease [74]. GBA activity reduces with aging [75], and GBA mutations may anticipate this reduction, reaching the critical threshold for substrate accumulation and subsequent impairment of a-synuclein traffic. The potential connection between α-synuclein accumulation and lysosome includes the disruption of the autophagy–lysosome system (ALS) and ubiquitin–proteasome system (UPS), compromising the degradation of misfolded proteins, partly contributing to neurotoxicity and neurodegeneration in PD patients [76]. Moreover, the presence of ALS-associated neuropathology and axonal degeneration has been found in the brains of a-GAL A-deficient mice [50]. The plausibility of the association of deficient activity of a-GAL A in AFD with a cascade mechanism at least partially involving the ones described in GBA-mutated subjects is illustrated in Figure 1. The illustrated lysosomal enzymatic pathways show that a-GAL A is involved in the production of glucocerebroside, and its deficiency may reduce glucocerebroside levels and therefore the substrate for GBA.

Few reports analyzed the enzymatic activity levels of a-Gal A in the peripheral blood and brain tissue of PD patients, often within a more extensive quantification of lysosomal enzymes’ activity. Wu et al. [49] measured GLA activity in leukocytes of 38 PD cases vs. 258 controls, finding lower GLA activity in PD patients (18.56 ± 1.49 units versus 22.24 ± 0.60 units, *p* < 0.05). Pchelina et al. [77] measured GLA activity in 42 Gaucher patients, 21 GBA/PD patients, 84 sporadic PD patients, and 62 controls, finding reduced GLA activity in GBA/PD patients (*p* = 0.001) and in Gaucher patients (*p* = 0.019), and a non-significant trend among sporadic PD patients (*p* = 0.178). Alcalay et al. [78] used dried blood spots of 648 PD cases vs. 317 controls to test whether reduction in lysosomal enzymatic activity in PD is specific to GBA, measuring the activity of GBA, acid sphingomyelinase (Niemann–Pick disease types A and B), a-GAL A galactosidase A (AFD), acid alpha-glucosidase (Pompe disease), and galactosylceramidase (Krabbe disease). They found that a-GAL A activity was lower in PD cases compared to controls, even excluding all GBA and LRRK2 G2019S carriers and early onset PD cases (2.85 μmol/L/h versus 3.12 μmol/L/h, *p* = 0.018); this difference was significant after stratifying by sex too. Using a regression model, higher a-GAL A activity was associated with lower odds of PD (OR = 0.54; 95% CI: 0.31–0.95; *p* = 0.032). The association raised in the study between lower GLA activity and PD status is even stronger than the association between lower GBA activity and PD. This observation adds fuel to the fire of the unresolved question on the specificity of the association between GBA mutations and PD vs. the wider contribution of several enzymes in the lysosomal pathways. The fact that GBA heterozygous mutations are by far more frequent (and not life-threatening) than the mutations in other lysosomal genes may overshadow this association, hiding a more complex and subtle combination of deficits and mechanisms. For example, the E326K variant is recorded in up to 3% of all healthy Caucasians [79], while even pathogenic mutations are not rare in selected populations (6–7% of all healthy Ashkenazi Jews) [80]. In this study, reduced GLA activity was associated with PD status when LRRK2 and GBA mutation carriers were included, and this association was even stronger in non-carriers of both sexes but significantly in women only. Lysosomal enzymatic activities have been measured and compared between PD and controls not only in peripheral blood but also in brain tissue and CSF. Nelson et al. [81] showed that a-Gal A enzymatic activity was significantly reduced in the postmortem brains of 10 subjects with advanced PD. In a second postmortem study [82] on two independent PD cohorts (n = 18 and n = 20), the authors postulated an association between sporadic PD and reduced activity of several lysosomal hydrolases, including a-Gal A, in the SN. Unfortunately, GLA was never reported in studies on lysosomal enzymes in CSF [83,84,85,86].

Therefore, the most convincing hypothesis is that GLA may have a potential independent role in PD, in addition to GBA, but maybe through different pathways of brain damage and different phenotypes.

In Table 1 are summarized the main levels of a plausible association between AFD and PD.

## 4. Research Perspectives and Therapeutic Implications

The association between AFD and PD is intriguing, but some issues deserve attention. PD is a disease with a high prevalence in older adults and a peak incidence of onset between 70 and 79 years, so many classical AFD patients may not live long enough to develop PD [86], but considering the improved life expectancy in the ERT era and the increasing diagnosis of late-phenotype AFD patients, it is possible that in the future a clearer association could emerge. Another limitation is that the cerebrovascular comorbidity in AFD patients might cause an extrapyramidal phenotype that is not purely neurodegenerative but secondary as a vascular Parkinsonism. The exact mechanisms of CNS involvement in AFD are not yet clearly understood. Several hypotheses have been proposed, including a complex and dynamic interplay between local hypoperfusion and neuronal Gb3 storage resulting in progressive atrophy of tissues and death of neurons according to a multiple-hits neurodegenerative model.

Moreover, the PD phenotype in AFD in advanced neuroimaging studies is very similar to the selective SN dysfunction described in early PD patients. A direct and prospective comparison between these two populations could help to explain the pathophysiology and clinical course of patients and to define the therapeutic strategy. In the study of Lohle et al. [18], patients with AFD showed slower hand and gait speed but did not manifest non-motor symptoms such as hyposmia or REM sleep behavior disorder, which are often present in idiopathic PD and in GBA-PD. The most prevalent theory is that the neurodegenerative prodromal phenotype of AFD has a main-motor extrapyramidal expression, and in isolated cases a PD-like phenotype is evident, with more prevalent axial symptoms and low effect to dopaminergic treatment. Advanced neuroimaging and post-mortem studies, even limited in number and reproducibility, support this hypothesis with mechanisms different from the ones outlined in sporadic PD and in GBA-PD. In patients with classical AFD and/or moderate heart and kidney involvement and a moderate to severe burden of cerebrovascular lesions, the vascular and neurodegenerative phenotypes may combine with, interplay with, and potentiate each other.

The future lines of research are: the improvement of the understanding of the biological mechanisms underlying neurodegeneration in AFD patients (deposition of Gb3 versus cerebrovascular lesions in the nigrostriatal network); the existence of a compensatory mechanism of increased a-GAL A activity upon PD neurodegeneration, as observed in LRRK2 carriers; the prospective multimodal comparison between AFD and sporadic PD patients; and finally the role of ERT chaperone therapy in neurodegeneration in AFD, remembering that ERT does not penetrate the blood–brain barrier.

## 5. Conclusions

AFD is a rare lysosomal storage disease whose neurological manifestations include a neurodegenerative pattern produced through several mechanisms. The main degenerative pattern derives from the motor system’s involvement with an extrapyramidal phenotype similar to sporadic PD but with low sensitivity to dopaminergic treatment. More studies are needed to assess the molecular and biological mechanisms of brain damage in AFD with a predilection for the extrapyramidal network.

## Figures and Tables

**Figure 1 biomedicines-10-03132-f001:**
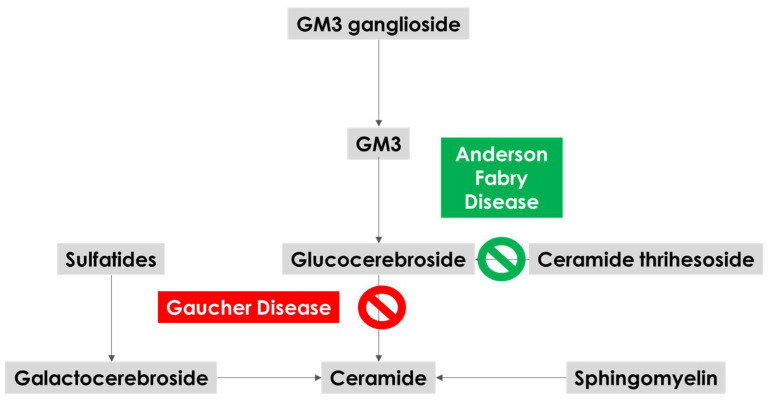
Schematic view of the main lysosomal enzymatic pathways affected by deficit activity of a-GAL A (green) and GBA (red).

**Table 1 biomedicines-10-03132-t001:** Summary of the levels of an association between AFD and extrapyramidal phenotype.

Level	Mechanism
Biological	Reduced a-GAL A activity in brain samples of PD patients.
Histopathological	Neurodegenerative hallmarks in few brain autopsies of AFD patients with concomitant extrapyramidal manifestations during life. Signs of substrate accumulation in brain areas crucial for extrapyramidal pathways in brain autopsies of AFD patients.
Clinical	Neurodegenerative prodromal phenotype in AFD patients.
Neuroradiological	MRI markers of neurodegenerative disease and extrapyramidal dysfunction in selected AFD patients’ subgroups.

## Data Availability

Not applicable.

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
