# Peer review of "Anderson–Fabry Disease: A New Piece of the Lysosomal Puzzle in Parkinson Disease?"

_biomedicines, 2022, doi:10.3390/biomedicines10123132_

Round 1
Reviewer 1 Report
Thank you for writing a the great review - I expect this to be a great help for those needing an update on the field, both from methodological and medical point of view.
Author Response
Many thanks fro your appreciation of our work. We added also a graphical abstract, as requested by the academic editor.
Reviewer 2 Report
Dear the EditorZedde M et al are proposing a novel idea for the potential linkage of AFD and PD. Although this hypothesis seems to be intriguing, these authors appear to only fucus on their own opinion. The reason is that AFD is one of frequently appeared LSD, while their basis of idea only derived from two cases (refs 53, 54). The subsequent description for molecular basis does not appear to directly explain the relationship between AFD and PD.Minor suggestion:1) It would be helpful to provide a graphical abstract of this review manuscript for readers of the Biomedicineds journal.2) Submission of this manuscript to any more neuroscience-specific journal would be recommended.
Author Response
First of all, we would like to thank the reviewer for reading our paper and for the suggestions.
Zedde M et al are proposing a novel idea for the potential linkage of AFD and PD. Although this hypothesis seems to be intriguing, these authors appear to only fucus on their own opinion. The reason is that AFD is one of frequently appeared LSD, while their basis of idea only derived from two cases (refs 53, 54). The subsequent description for molecular basis does not appear to directly explain the relationship between AFD and PD.
The issue raised by the reviewer is a good point. In the general field of LSD the neurological involvement of AFD is usually focused on cerebrovascular manifestations (in the CNS) and do not include neurodegenerative clinical and neuorimaging features. This "dark" side of the disease is increasingly gaining interest in the neurlogical field, both starting from the movement disorders specialist and from the rare neurovascular disorders specialist. The aim of our paper is to present this issue to a general audience. It is still a working hypotheses with missing points and building evidence, starting from the biological mechanisms and histhopathological proofs and the two references mentioned by the reviewer provide some clues on this association using the gold standard for CNS diseases diagnosis. There are only few reports on brain autoptic findings in AFD patients and we are not able to retrieve more information but the scarse available data have been proposed in building an hypothesis to be proven in the future.
About the general biological mechanisms we added the figure 1 and the following sentences in the text.
"The plausibility of the association of deficient activity of a-GAL-A in AFD with a cascade mechanism al lest partially involving the ones described in GBA mutated subjects is illustrated in figure 1. The illustrated lysosomal enzimatic pathways show that a-GAL-A is involved in the production of glucocerebroside and its deficiency may reduce the glucocerebroside levels and therefore the substrate for GBA."
Minor suggestion:1) It would be helpful to provide a graphical abstract of this review manuscript for readers of the Biomedicineds journal.
We will address this issue after completing the revision of the manuscript according to the reviers' comments and suggestion, if the Editor agrees.
2) Submission of this manuscript to any more neuroscience-specific journal would be recommended.
Thanks for the suggestion. We aimed to share the intriguing hypothesis of a neurodegenerative side of AFD with the wide communiti od LSD specialist as first step. We will consider this suggestion according to the Editor's choices.

Reviewer 3 Report
The manuscript entitled “Anderson-Fabry Disease: a new piece of the lysosomal puzzle in Parkinson Disease?“ summarizes the clinical and neuroimaging observations supporting the association of Anderson Fabry Disease (AFD) with a clinical phenotype of Parkinson Disease (PD). The subject is extensively covered, critically commented, and supported by a number of updated references. The manuscript is overall balanced and well-written but should undergo minor proofreading.
Minor:
Line 124: please correct “pathopgenesis” in “pathogenesis”
Line 211-212: the sentence “lysosomal integral membrane protein type B (LIMP-2), which is the glucocerebrosidase chaperone [49], which encodes the glucocerebrosidase chaperone LIMP-2” it is not clear and needs to be modified.
Line 213: replace GCA with GCase
Line 320: “the excluding” should be deleted
Line 329: please correct “recuction” in “reduction”
Line 333: please correct “contributting” in “contributing”
Line 339: please correct “alower” in “a lower”
Line 352: please correct “evfen” in “even”
Line 353: “the” is repeated, please delete one
Line 354-355: “GBA” is repeated, please delete one
Line 357: does authors mean “other lysosomal genes may overlook…”?
Line 358: please correct “od” in “of”
Finally, I would suggest that the authors include a summarizing figure or outline.
Author Response
First of all, we would like to thank the reviewer for his/her appreciation of our paper.
We addressed the mentioned issues, and in particular we changed the sentence in lines 211-212 as follows:
"The deficiency of other lysosomal enzymes is also associated with extrapyramidal disorders in general and PD in detail. Some examples of this association are Sphingomyelin Phosphodiesterase 1 (SMPD1) deficit causing Niemann-Pick A [46-48] and the deficit of lysosomal integral membrane protein type B (LIMP-2), which is the glucocerebrosidase chaperone [49]."
Moreover, we added a figure (figure 1) with this caption:
Figure 1. Schematic view of the main lysosomal enzymatic pathways affected by deficit activity of a-GAL-A (green) and GBA (red).
and the following sentences in the text:
"The plausibility of the association of deficient activity of a-GAL-A in AFD with a cascade mechanism al lest partially involving the ones described in GBA mutated subjects is illustrated in figure 1. The illustrated lysosomal enzimatic pathways show that a-GAL-A is involved in the production of glucocerebroside and its deficiency may reduce the glucocerebroside levels and therefore the substrate for GBA."
We also added a table summarizing the main levels of a lasible association between AFD and PD, as follows:
In table 1 are summarized the main levels of a plasible association between AFD and PD.
Table 1. Summary of the levels of an association between AFD and extrapyramidal phenotype.
|
Level |
Mechanism |
|
Biological |
Reduced a-GAL-A activity in brain samples of PD patients |
|
Histopathological |
Neurodegenerative hallmarks in few brain autopsies of AFD patients with concomitant extrapyramidal manifestations during life Signs of substrate accumulation in brain areas crucial for the extrapyramidal pathways in brain autopsies of AFD patients |
|
Clinical |
Neurodegenerative prodromal phenotype in AFD patients |
|
Neuroradiological |
MRI markers of neurodegenerative disease and extrapyramidal dysfunction in selected AFD patients subgroups
|

Round 2
Reviewer 2 Report
Dear the Editor
The Authors have provided the resonable response to each concern from the Reviewer. It seemed conceivable to positively consider this idea as a working hypothesis at this stage. Fig. 1 could be suitable for the Graphical abstract of this paper.
Author Response
Many thanks for your suggestions. We incorporated figure 1 in the graphical abstract.